# Recent Developments in Activated Carbon Catalysts Based on Pore Size Regulation in the Application of Catalytic Ozonation

Jin Yang [1,2], Liya Fu [1], Fachao Wu [3], Xingxing Chen [1,4], Changyong Wu [1,5,*] and Qibao Wang [2,*]

1   Research Center of Environmental Pollution Control Engineering Technology, Chinese Research Academy of Environment Sciences, Beijing 100012, China
2   School of Chemical & Environmental Engineering, China University of Mining and Technology (Beijing), Beijing 100083, China
3   Hebei Key Laboratory of Hazardous Chemicals Safety and Control Technology, School of Chemical and Safety, North China Institute of Science and Technology, Langfang 065201, China
4   College of Architecture and Environment, Sichuan University, Chengdu 610065, China
5   State Key Laboratory of Environmental Criteria and Risk Assessment, Chinese Research Academy of Environmental Sciences, Beijing 100012, China
*   Correspondence: changyongwu@126.com (C.W.); wqb@cumtb.edu.cn (Q.W.)

**Abstract:** Due to its highly developed pore structure and large specific surface area, activated carbon is often used as a catalyst or catalyst carrier in catalytic ozonation. Although the pore structure of activated carbon plays a significant role in the treatment of wastewater and the mass transfer of ozone molecules, the effect is complicated and unclear. Because different application scenarios require catalysts with different pore structures, catalysts with appropriate pore structure characteristics should be developed. In this review, we systematically summarized the current adjustment methods for the pore structure of activated carbon, including raw material, carbonization, activation, modification, and loading. Then, based on the brief introduction of the application of activated carbon in catalytic ozonation, the effects of pore structure on catalytic ozonation and mass transfer are reviewed. Furthermore, we proposed that the effect of pore structure is mainly to provide catalytic active sites, promote free radical generation, and reduce mass transfer resistance. Therefore, large external surface area and reasonable pore size distribution are conducive to catalytic ozonation and mass transfer.

**Keywords:** activated carbon; catalytic ozonation; mass transfer; pore structure; external specific surface area

## 1. Introduction

Ozone is an effective oxidant with extremely high redox potential ($E^0 = 2.07$ V), and it is widely applied in all kinds of wastewater treatment [1–3]. Based on ozone, a series of advanced oxidation processes (AOPs) have been developed, and heterogeneous catalytic ozonation is a promising technology because free radicals (mainly hydroxyl radicals) are generated in the reaction process, which can quickly oxidize various organic pollutants with non-selectivity, and completely mineralize micropollutants and natural organic pollutants without secondary pollution [1,4–17].

Activated carbon and metal oxides are the most commonly used catalysts in heterogeneous catalytic ozonation [18,19]. Compared with metal oxide catalysts (such as alumina-based catalysts), activated carbon catalysts have a more developed pore structure, larger specific surface area, more active sites, and better uniform dispersion of active components on the surface. These advantages make activated carbon a more effective catalyst or catalyst carrier in catalytic ozonation [1,6,7,18,20–23]. For catalysts, the degradation efficiency of target pollutants will be greatly improved when the specific surface area and the possible active sites increase [24]. At the same time, the increase of specific

surface area will also enhance the adsorption capacity of the catalyst. The strong adsorption capacity and efficient catalytic ability jointly promote the degradation of the target pollutant, which shows the great advantage of activated carbon catalysts. Because different wastewaters often require different catalysts, it is necessary to develop activated carbon catalysts with appropriate pore structures according to the characteristics of the wastewater. The directional preparation of activated carbon catalysts is also one of the current research hotspots [22,25–27].

Currently, in heterogeneous catalytic ozonation, activated carbon is usually used as a catalyst or catalyst carrier [5]. The influencing factors of the wastewater treatment process are mainly divided into operation parameters and catalyst properties [28,29]. The operating parameters mainly refer to the initial solution pH, catalyst dosage, ozone dosage, reaction temperature, initial ozone concentration, etc. [30]. The catalysts' properties mainly refer to the pore structure (i.e., the pore shape, pore volume, average pore size, and pore size distribution) and the surface properties (i.e., surface morphology and functional groups) [31]. Differences in pore structure and surface properties of catalysts will affect the number of active sites, the dispersion of active components, and the mass transfer of ozone molecules, thus directly affecting the degradation efficiency [32]. Yu et al. [33] stated that porosity and carbon configuration were two key factors of biomass-based activated carbon in the advanced oxidation process.

However, most researchers have not emphasized the preparation of activated carbon, often using commercial activated carbon or simply adjusting the pore structure and surface properties of activated carbon by modification [34–37]. There are also many new types of structures of activated carbon catalysts that have been developed. But few studies focus on the preparation of activated carbon catalysts with appropriate pore structures for specific wastewater treatment [8,38,39]. Moreover, even these studies have rarely established the quantitative relationship between the pore structure of activated carbon catalysts and catalytic activity. Most of the studies are qualitative or descriptive, that is, they conclude that a change in the specific surface area or pore structure of the catalyst will lead to a change in catalytic activity. There is no in-depth discussion and no single activated carbon catalyst that is suitable for all wastewater qualities [21].

According to the different raw materials and methods, the preparation process of activated carbon catalyst is always different, but the main preparation steps are the same: raw material–carbonization–activation–load [40]. The preparation process is mainly carried out around the pore structure, specific surface area, and intensity of activated carbon, so theoretically every step has an impact on the pore structure of the catalyst, especially the raw material, production conditions, and activation parameters [40–45].

In this paper, recent developments in pore size regulation of activated carbon catalysts and their application in wastewater treatment are reviewed, which provides a reference for the directional preparation of activated carbon catalysts and further research on the effect of pore size on catalytic ozonation and mass transfer.

## 2. Application of Activated Carbon in Catalytic Ozonation

### 2.1. Brief Introduction of Activated Carbon in Catalytic Ozonation

Various studies have shown that the presence of activated carbon can improve the degradation efficiency of wastewater and target pollutants, which is generally higher than that of other catalysts supported on metal oxides or zeolite [1,38,46–50]. In this catalytic ozonation process, activated carbon behaves as an adsorbent, catalyst, or initiator of free radicals produced by ozone decomposition [1,4,5,8,10,35,38,46,51–56]. The contribution of activated carbon as a catalyst or catalyst carrier can be possibly ascribed to the following aspects [46,57,58]: (1) activated carbon provides a high specific surface area, and both organic compounds and ozone molecules can be adsorbed and reacted on it; (2) activated carbon accelerates ozone decomposition and produces a large number of free radicals, mainly hydroxyl radicals ($\bullet$OH); (3) ozone reacts with surface groups of activated carbon

and generates adsorbed $H_2O_2$, which reacts with ozone in the bulk solution to produce hydroxyl radicals.

However, although many experiments and studies have confirmed that hydroxyl radicals produced by ozone decomposition are the main reactive oxygen species (ROS), many studies have also proved that there are other ROS, such as superoxide radicals ($O_2^{\bullet-}$), singlet oxygen ($^1O_2$), and hydrogen peroxide ($H_2O_2$), and some researchers have even reported that there was no generation of hydroxyl radicals [1,2,10,53,56,59–67]. In addition, it is unclear whether hydroxyl radicals are generated on the surface of activated carbon, boundary layer, or in the bulk solution, which may be related to the properties of activated carbon and target pollutants, and solution pH [4,5,39,46,51,68]. Faria et al. [69] reported the degradation of oxamic and oxalic acids by activated carbon and discovered that when pH was 3, the reaction mainly occurred on the surface of the activated carbon; when pH was 7, the reaction took place on the surface of activated carbon and the bulk solution. A similar finding was also reported by Wang et al. [4]. However, Beltran et al. [70] studied the kinetics of ozone decomposition and confirmed that hydroxyl radicals were mainly formed in the bulk solution rather than the surface interaction. It should be noted that the generation of hydroxyl radicals is certainly related to the active sites on the surface of activated carbon catalysts [1,51,57]. It is also certain that adsorption plays a very important role, as either ozone molecules or pollutants are adsorbed on the surface of activated carbon, otherwise it will not have a catalytic effect [51,57].

Researchers have adopted various reaction mechanisms, in which activated carbon was a catalyst or catalyst carrier. Faria et al. [69] proposed two possible ozonation reaction mechanisms when studying the oxidation of oxalic acid with activated carbon. The first one is that ozone molecules decompose on the surface of activated carbon to produce free radicals, such as hydroxyl radicals, and then hydroxyl radicals oxidize the pollutants (R). This is the •OH mechanism, one of the two representative mechanisms for the heterogeneous catalytic ozonation; catalysts (especially metal or metal oxide, such as aluminum oxides, iron oxides, manganese oxides, etc.) can increase the solubility of ozone and initiate ozone decomposition to generate •OH [71].

$$O_3 \xrightarrow{AC} HO\bullet \tag{1}$$

$$HO\bullet + R \rightarrow P \tag{2}$$

The second mechanism is that ozone molecules are absorbed on the surface of activated carbon to produce surface oxygen-containing active species (AC-O, including HO•), which react with adsorbed organic molecules.

$$O_3 + AC \rightarrow AC - O \tag{3}$$

$$R + AC \rightarrow AC - R \tag{4}$$

$$AC - R + AC - O \rightarrow P \tag{5}$$

In another paper, Faria et al. [72] illustrated the main possible reaction pathways occurring in the ozonation of dyes catalyzed by activated carbon, as shown in Figure 1. In this process, there exists an interfacial reaction mechanism, and the main function of catalysts is to act as adsorptive material, so the catalysts always have a large surface area and highly developed pore structure, as in the case of activated carbon, zeolite, honeycomb ceramics, etc. [71]. Although zeolite plays an extremely important role in adsorption and catalysis, its well-ordered and rigid structure is very different from activated carbon, and the specific catalytic mechanism is also different [73].

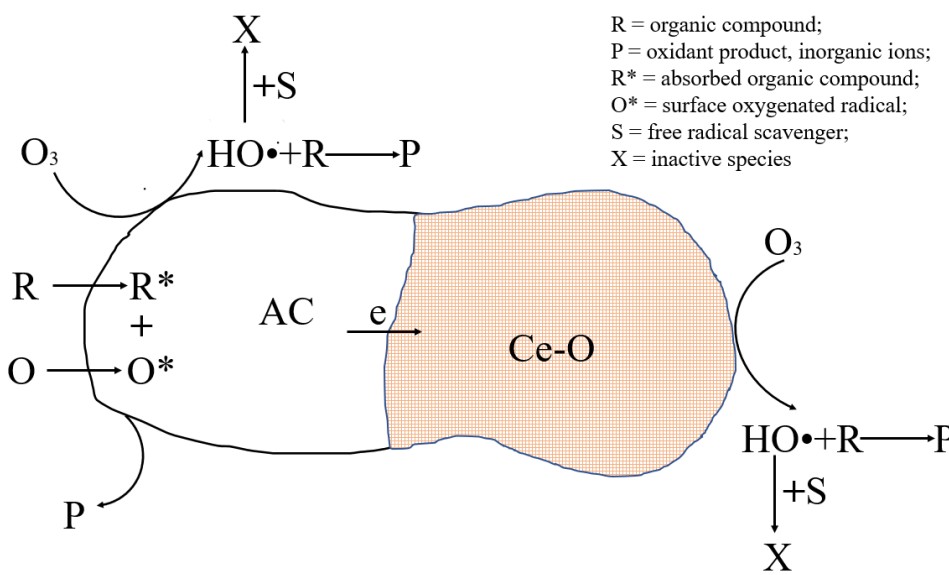

**Figure 1.** Illustrative representation of the main reaction pathways occurring during ozonation catalyzed by AC-Ce-O (based on [72]).

Hao et al. [7] proposed a new reaction mechanism when studying the catalytic ozonation of polycyclic aromatic hydrocarbons (PAHs) with cerium supported on activated carbon (Ce-AC). The reaction Equation is (6)–(10), and the proposed reaction pathway is shown in Figure 2.

$$P + AC \rightarrow P_{in} - AC \tag{6}$$

$$O_3 + AC \rightarrow O_{3in} - AC \tag{7}$$

$$P_{in} - AC + O_{3in} - AC \rightarrow P_0 + H_2O_2 \tag{8}$$

$$H_2O_2 + AC \rightarrow H_2O_2 - AC \tag{9}$$

$$H_2O_2 - AC + O_3 - AC \rightarrow \bullet OH + AC \tag{10}$$

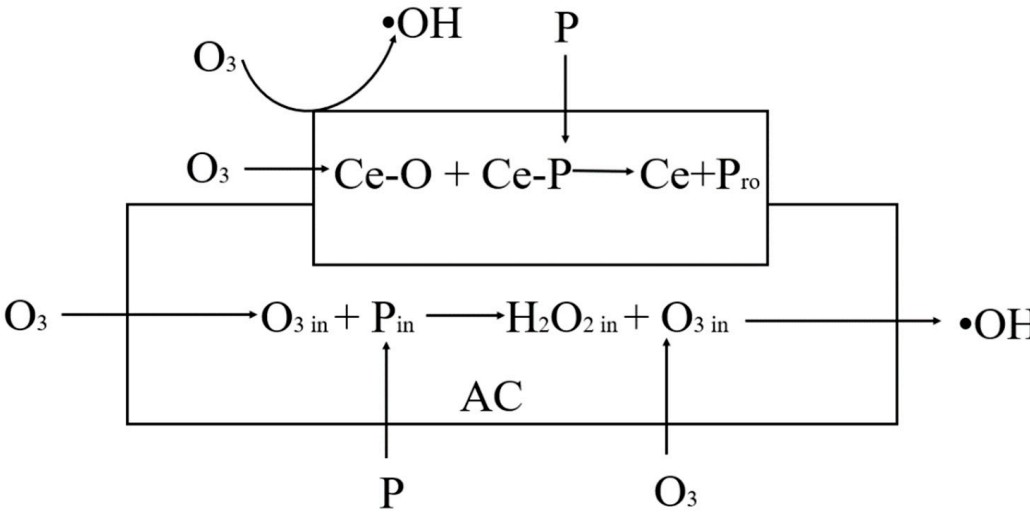

**Figure 2.** The proposed reaction pathway of catalytic ozonation by Ce-AC (based on [7]).

In addition, on the basis of summarizing previous studies, Huang et al. [74] removed aqueous oxalic acid with MnOx-activated carbon and proposed a possible reaction mechanism, as shown in Figure 3.

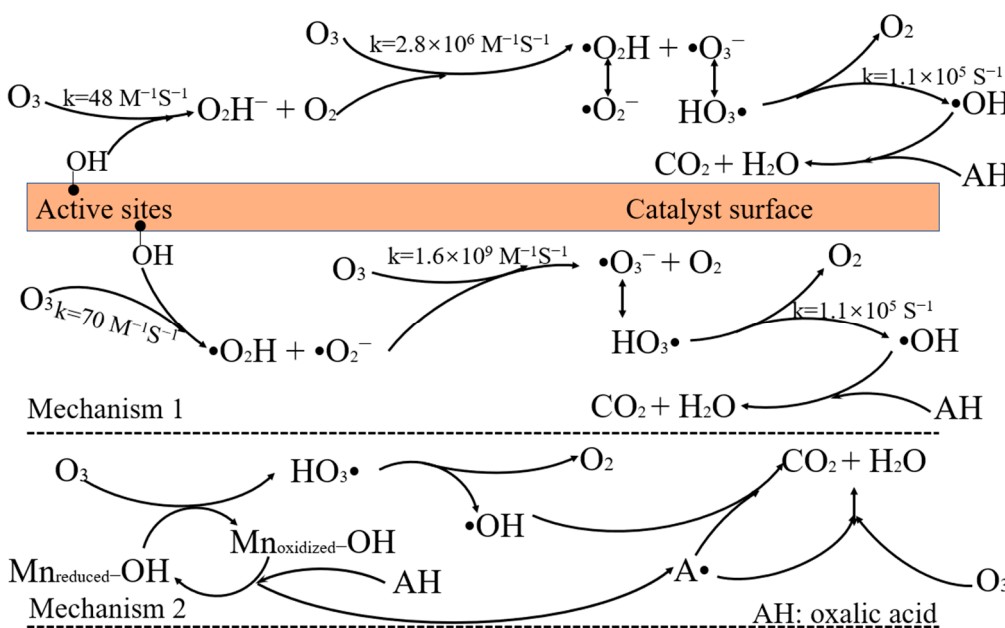

**Figure 3.** Mechanism of heterogeneous catalytic ozonation with $MnO_x/SAC$ (based on [74]).

It is worth noting that hydrogen peroxide ($H_2O_2$) may be generated in the AC catalyst/ozone system. As the most practical oxidizing agent, $H_2O_2$ has a certain oxidation ability, and it can react with metals (such as Al, Fe, etc.) to generate hydroxyl radicals, which can oxidize pollutants or further react with $H_2O_2$ to generate $\bullet OH_2$ [75,76]. However, $H_2O_2$ is a weak acid, which has lower oxidant potential ($E^0 = 1.77$ V) than ozone ($E^0 = 2.07$ V), and it alone does not lead to the generation of $\bullet OH$. In addition, it can act as a radicals scavenger, and these characteristics will decrease the degradation efficiency; therefore, the ozone-based AOPs have higher treatment efficiency [71,77]. Fu et al. [78] believed that in the ozone/$H_2O_2$ system, $H_2O_2$ will accelerate the $\bullet OH$ yield in the ozone transformation process. However, the oxidation effect of this ozone/$H_2O_2$ system is not steady, and it has limited advantages in TOC removal compared with the single ozone system.

However, although researchers have proposed many mechanisms to explain the reaction process, these are contradictory, and the mechanism is still unclear. According to the report of Nawrocki et al. [51] and other researchers [67], the main reason is that these catalytic ozonation are very complicated processes, and different mechanisms are only appropriate for specific systems. In addition, these may be caused by experimental errors and/or analysis errors, such as the lack of accurate pH control, the amount and purity of catalysts, the impact of adsorption, water quality, inappropriate indicators, various parameters, and improper application and analysis of free radical quencher, etc. [51,57,59,61,63,65,67,79,80].

As mentioned above, activated carbon can greatly improve catalytic efficiency, but its application is limited due to the following problems: activated carbon powder is difficult to recover; it will be modified after long-term operation; it has high operating costs and poor stability, etc. [46,62,64,81]. The key to reducing cost is the regeneration of activated carbon, and there are various methods to achieve this, such as ultrasound, plasma, microwave, thermal, biological, chemical, and electrochemical methods, etc., the most common method being thermal regeneration [46]. In order to solve the problem of the difficult recovery of activated carbon catalyst, magnetization is a promising method, which has been studied by many researchers [56,82,83]. Lu et al. [84] prepared a series of novel ferromagnetic sludge-based activated carbon with a good magnetic ability by the coprecipitation method. Azam et al. [85] produced magnetic AC particles with the impregnation method.

## 2.2. pH

The initial solution pH was a key operational parameter affecting the catalytic process, and it will also change the surface characteristics of activated carbon catalyst [17,29,55,57,68,74,75,86–90]. The decomposition of ozone molecules is strongly dependent on the initial solution pH, which not only affects the degradation efficiency but may also change the catalytic mechanism [10,19,28,68,77,91]. Some researchers conclude that, at acid and near neutral pH, there are no hydroxyl radicals generated; when pH is 6~9, ozone decomposition and hydroxyl radicals were produced; and other researchers found that alkaline media is more conducive to the degradation of the pollutant [1,4,8,9,28,29,32,36,51,56,74,92,93]. However, Beltran et al. [70] proved that oxalic acid mineralization is carried out through the free radical pathway at pH 2.5. Liu et al. [47] treated reverse osmosis concentration with activated carbon and found that hydroxyl radicals were produced under alkaline conditions; however, the total organic carbon (TOC) removal rate at pH 9 was higher than that of at pH 11, because the carbonate in the solution quenched the hydroxyl radicals at pH 11. Fijołek et al. [28] believed that only under acidic conditions, ozone molecules decompose on the surface of activated carbon and produce a large amount of hydrogen peroxide. The catalytic activity reported in the literature for various catalysts is often different because pH is not exactly the same. However, even at the same pH, different catalysts might behave differently mainly due to different surface characteristics of activated carbon [80,86]. In addition, there are also some reports that the interaction of ozone with the carbon surface is not dependent on pH [37]. In conclusion, pH is an extremely important and sensitive parameter, which should be carefully controlled and monitored [67].

## 3. Pore Structure Regulation and Its Effect on Catalytic Ozonation

### 3.1. Overview of Pore Structure Regulation

The pore diameter of activated carbon is mainly divided into three types according to its size: micropore (pore diameter less than 2 nm), mesopore (pore diameter 2–50 nm), and macropore (pore diameter higher than 50 nm) [21,22,40,94–99]. The number of macropores is very small, and they lead directly to the outer surface of activated carbon and mainly act as a channel for the adsorption of molecules; the mesopore is a transition pore, and activated carbon as a catalyst carrier needs more mesopores; micropores are the most numerous and the major contributor to the specific surface area, but they need the channel function of macropores and the transition function of mesopores [21,22,31,40,96,100]. Different processing objects require activated carbon with different pore structures; therefore, the adjustment of the porosity and the pore size distribution of activated carbon can widen its applications [101]. Xie et al. [22] and Mansour et al. [31] believed that activated carbon with good adsorption performance needs fully developed micropores, as well as a certain number and shape of mesopores and macropores. In addition, activated carbon as a good catalyst carrier also needs an appropriate aperture number and ratio. Pore structure adjustment depended on the raw material and preparation process. In the carbonization step, volatile components in the raw materials escape under moderate temperature, and then the initial pore structure was formed. The initial pores are further expanded through various activation methods and the use of activation reagents at the activation step, and activated carbon with different pore sizes and distributions was obtained [21,22]. Due to the highly developed pore structure, activated carbon has a large specific surface area and surface Gibbs free energy, it can be used as an adsorbent with strong adsorption capacity or a catalyst carrier supporting various active components. The pore structure is one of the most important factors of activated carbon catalysts [1,21].

The pore structure of catalysts obtained from different raw materials, preparation methods, and preparation parameters are often different, resulting in different catalytic efficiency in the treatment of industrial wastewater. To improve the catalytic efficiency, activated carbon catalysts with suitable pore structures should be developed according to the characteristics of water quality [25].

### 3.2. The Regulation Methods and Influencing Factors of Pore Structure

3.2.1. Raw Material

The prepared activated carbon will always retain the structure characteristics of the raw material or precursor; in addition, the properties of raw materials significantly affect the performance and application of the final product [41–43,45,102,103]. Therefore, it is very important to select appropriate raw materials to prepare activated carbon catalysts. For instance, due to the numbers of rough hollow tubes and the special laminated structure of cotton, the specific surface area of cotton-based activated carbon is larger than activated carbon made from other biomasses [104]. The main component of activated carbon is carbon, and theoretically any material rich in amorphous carbon can be used for the preparation of activated carbon, such as coal (lignite, anthracite, etc.), wood, biomass (coconut shell, palm kernel shell, rice straw, nutshells, tea waste, etc.), petroleum, organic polymers (polystyrene), brewing yeast, and sludge, ion-exchange resins, etc.; among them, coal and biomass are the most commonly used raw materials [5,21,31,40–43,96,97,100,102,105–114]. Table 1 summarizes the various raw materials used to prepare activated carbon, and the characteristics of the raw materials and the prepared activated carbon. Some researchers have found that activated carbon derived from plastic waste has adequate micropores, and that activated carbon derived from biomasses has larger micropores and mesopores [31,115].

Coal is an important raw material for the preparation of activated carbon, because of the low cost and convenient adjustment of pore structure [31,105,107]. However, there are many impurities in coal, especially high ash content, and the pore structure is limited by the metamorphism degree and viscosity of coal itself, so the specific surface area of the coal-based activated carbon is relatively small. For example, the activated carbon made from anthracite developed micropores, while that made from lignite has more mesopores and macropores, and low strength. In order to make full use of the characteristics of different types of coals, the method of coal blending and adding binder can be used to prepare activated carbon [22,105]. Biomass, such as wood, walnut shell, nutshell, coconut shell, etc., is mainly composed of biopolymers (hemicellulose, cellulose and lignin), with fewer impurities and low ash. Therefore, activated carbon produced from biomass has developed microporous structure, large specific surface area and many functional groups [31,97,102,104,115–117].

The average particle size and particle size distribution of the raw material are also important factors. The smaller the particle size of the raw material and the more reasonable the particle size distribution, the more developed the pore structure of prepared activated carbon and the larger the specific surface area. The appropriate average particle size and particle size distribution have been selected according to the conditions [100,118].

3.2.2. Carbonization

Carbonization is the first stage of activation (especially the physical activation process): the raw material is carbonized in the absence of air and any chemicals [119,120]. Moisture, most of the organic composition, and volatile components in the carbonaceous material will be eliminated during the carbonization process at moderate temperature and inert atmosphere. Then, the initial pores are formed, and the carbon content increases, which is the basis for further activation [43,81,107]. There are three main forms of carbonization: pyrolysis, hydrothermal carbonization (HTC), and microwave-assisted carbonization [81]. The main factors of the carbonization process (taking pyrolysis as an example) are carbonization temperature and carbonization time; the heating rate of carbonization temperature has a limited effect on the pore structure of carbonized material [121,122].

The main purpose of the raw material selection and carbonization process is to form an activated precursor suitable for the pore structure development of activated carbon. At present, there is no consensus on the influence of the microcrystalline structure of carbonized materials on the pore structure of activated carbon. Although a lower degree of graphitization is beneficial to the activation, it is not necessarily beneficial to the formation of activated carbon with a large surface area [120]. The activation reaction is relatively

mild gasification in the reaction process. Although the energy consumption is high, the potential for micropore development is also greater. For coal-based activated carbon, the microcrystalline structure of carbonized materials is called "defective graphite layer", which is composed of the aromatic hydrocarbons in the coal coke macromolecular structure and the carbon atoms in the structure similar to the graphite microcrystalline structure. The disordered arrangement of the "defective graphite layer" is the basis for the formation of activated carbon pore structure [119,123].

Carbonization temperature. Carbonization temperature has a great influence on the formation of initial pores. Different carbonization temperatures will cause variations in specific surface areas and pore structures [124,125]. Carbonization temperature that is too high or too low is not conducive to the development of the pore structure of activated carbon [33]. With the increase in carbonization temperature, the initial pores increase, and when the carbonization temperature is too high, the micropores will collapse and the polycondensation reaction will result in the increase of mesopores [116]. Some researchers have found that the increase in carbonization temperature will lead to an increase in surface area [33,44,56,116]. Zhao et al. [124] and Tabak et al. [115] found that the sample with a carbonization temperature of 900 °C had the maximum specific surface area compared with the carbonization temperature of 800 °C and 850 °C. The higher carbonization temperature is favorable for increasing the surface area and pore volume of activated carbon. However, Allwar et al. [119] found that when carbonization increased from 800 °C to 900 °C, the surface area sharply decreased from 2301 $m^2/g$ to 1707 $m^2/g$. The appropriate carbonization temperature should be selected according to the characteristics of the raw materials and the pore structure requirements of activated carbon.

### 3.2.3. Activation

The activation process is the core stage of the preparation of activated carbon, which is the further expansion of the pore structure of the precursor. The activation process involves three major stages: elimination of tarry substances, burning the elementary carbon crystal, and oxidation of the carbon particles [81]. The main factors are the type of activation, activator, activation temperature, impregnation ratio, activation time, and gas partial pressure, etc. [41,100,102,121]. Activation methods can be divided into physical activation, chemical activation, physicochemical coupling activation method, microwave activation, template method, etc.; the most commonly used methods are physical activation and chemical activation [21,31,43,94,97,101,107,112,122,126–130]. The preparation of activated carbon derived from physical activation and chemical activation is shown in Figure 4, and the effect of activation on the properties of activated carbon is displayed in Table 1.

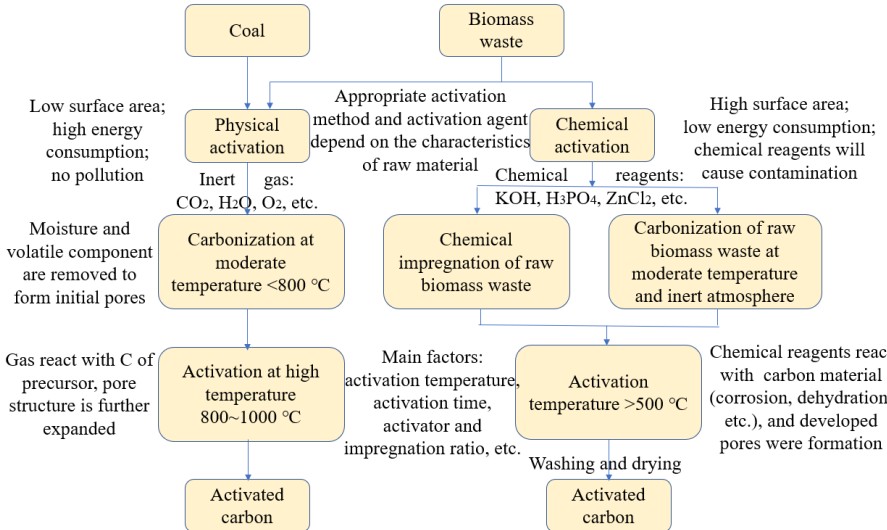

**Figure 4.** The production of activated carbon by physical and chemical activation (based on [97]).

Physical Activation

In the case of physical activation, water vapor (800~1000 °C), carbon dioxide (800~1000 °C), oxygen (below 800 °C), flue gas, and air are used as activators [43,95,96,101,107,131,132]. The essence of the physical activation is the redox reaction between the gas activator and the carbon atom in graphite microcrystalline [103]. The gasifier molecule makes contact with the internal carbon atom and reacts with it at high temperature. The gas will corrode and burn the surface of the carbonized material and remove some uncarbonized materials that block pores. The combination of the two processes can produce activated carbon with developed pore structure and large specific surface area [107]. Chen et al. [95] believed that the activation of multi-walled carbon nanotubes by $CO_2$ and air is a gas-solid reaction process, which is related with the reactivity and diffusion rate of gases. Because the reaction rate of carbon with $CO_2$ is much slower than that of oxygen in the air, more micropores will be formed by the activation with carbon dioxide. The main influence factors are activation temperature, activators, and the reactivity of carbonaceous materials [40,43,102]. The higher the activity of the gasification agent, the wider of the pore size distribution and the higher the mesopore proportion of activated carbon (for example, with oxygen as the activator); while the lower activity of the gasification agent, the better the development of micropores or ultra-micropores structure (for example, with carbon dioxide as the activator) [43,102,127,133]. However, the specific surface area of activated carbon produced by physical activation is rarely higher than 2000 $m^2/g$ and the yield is relatively low [43].

The control methods of pore structure adjustment can be summarized as reaction rate control and burn-off degree control. Reaction rate control means the increasing or decreasing the difference of activation reactivity between different regions of the precursor in the reaction process by changing the crystal structure of the precursor and the type of reaction gas. This difference in reactivity will change the ratio of "pore making" and "pore expanding" in the activation process, and then the purpose of pore structure regulation will be achieved [123]. Under low activation temperature and high gas partial pressure, the reaction rate is relatively low and the pore expanding effect is obvious, meaning that the prepared activated carbon has developed mesopores; on the other hand, under high temperature and low gas partial pressure, the prepared activated carbon develops micropores [44,123,134,135]

The burn-off degree control means that activated carbon with different pore structures can be obtained by prolonging or shortening the activation reaction time under the condition of determining the activation temperature and partial pressure of the activated gas [123,132]. The whole capacity of activated carbon increases with the increase in burn-off degree, and peaks occur between 50% and 80%, depending on the raw materials and activation conditions. The burn-off degree can reflect the full development of activated carbon pore structure to a certain extent. With the increase in burn-off degree, the pore structure of activated carbon will experience three stages: development, expansion, and collapse. In the development stage, a new pore structure is generated constantly and the specific surface area of activated carbon increases; at the expansion stage, the specific surface area of activated carbon is relatively stable, but the pore volume constantly expands and the mesopore content increases; in the collapse stage, the pore structure of activated carbon changes and the specific surface area decreases significantly [22,44,123,134,135]. Moreno-Piraján et al. [107] prepared activated carbon with carbon dioxide as an activator, and found that when the burn-off increased from 8.3% to 43.20%, the specific surface area, micropores volume, and total pore volume increased; similar results were obtained when other activators were selected.

Chemical Activation

Chemical activation often uses phosphoric acid ($H_3PO_4$), potassium hydroxide (KOH), zinc chloride ($ZnCl_2$), and other chemical reagents as activators, which were mixed with the raw materials or chars in a certain proportion to prepare activated carbon [5,22,43,85,101,104,107,116,127,128,132,135,136]. The chemical reagents enter the

interior of the carbon materials, and a series of reactions such as corrosion, dehydration, moistening, and aromatic synthesis occur in the interior of the carbon, until finally activated carbon with a developed pore structure is formed [137]. Some researchers believe that the use of chemical activators will weaken or break the chemical bonds between lignocellulosic substances in biochar, convert them into smaller organic fractions that are easy to remove in the activation step, and activated carbon with a well-developed porous structure will form [127]. However, the mechanism of the activation process is not very clear [138]. Studies have shown that the activated carbon prepared by potassium hydroxide has high micropore volume and narrow pore size distribution, while that prepared by phosphoric acid has high macropore volume and broader pore size distribution [127]. This result was in accordance with the research of Chen et al. [95]. In addition, the most important use of phosphoric acid as an activator is to increase the acidic functional groups on the surface of activated carbon [116]. Due to the corrosion, danger, and high-cost disadvantages of using potassium hydroxide or sodium hydroxide as activators, some researchers have used milder activators such as potassium carbonate ($K_2CO_3$) and potassium chloride (KCl) [116]. Moreno-Piraján et al. [107] found that thermal activation of phosphoric acid-activated samples showed a higher rate of activation compared to the activation using potassium chloride. Li et al. [116] concluded that the activity order of four activators from lowest to highest was potassium hydroxide, potassium carbonate, phosphoric acid, and zinc chloride. Rashidi et al. [96] found that, compared to phosphoric acid, carbon activated by potassium hydroxide has a larger specific surface area and higher micro–mesopore volume. Although chemical activation has the advantages of a large surface area, low activation temperature, short heat treatment time, and high carbon yield, it also has disadvantages such as high reagent cost, additional washing operation, and residual problems, especially the residual chemical reagents, which may act as catalysts for oxidation and will reduce the number of active sites, and affect the rate of activation [31,40,43,102,107,121,138,139].

Compared with physical activation, chemical activation can promote the development of a porous structure and produce activated carbon with a larger specific surface area, lower sulfur, and lower ash; it also can reduce activation temperature, activation time, and the formation of tar, but the use of chemical reagents can cause environmental pollution and equipment corrosion problems [43,44,101,102,106,110,119,122,135]. Different activation methods and activators should be selected according to the characteristics of the precursor and the intended use of the derived AC [81,97]. Coal-based activated carbon is usually produced by the physical activation method, because it is easy to produce activated carbon with developed micropores, and the reaction is safer than chemical activation. Phosphoric acid and zinc chloride in the chemical activation method are not suitable for activating coal-based activated carbon; potassium hydroxide also easily destroys the microcrystalline structure of coal char, therefore affecting the pore size and strength of activated carbon. Biomass-based activated carbon usually adopts the chemical activation method; the pore structure of prepared activated carbon was well developed, the specific surface area was huge, the pore size and pore diameter distribution were easy to control [22,102,106,119]. Nowicki et al. [113] prepared activated carbon from four different raw materials (corn cobs, cherry stones, coffee, and tobacco industry waste materials), and concluded that activated carbon with the largest specific surface area was obtained by KOH activation rather than physical or direct activation.

**Table 1.** Effects of raw materials and activation method on the properties of activated carbon.

| Raw Material | Raw Material Characteristics | Activation Agent and Main Parameters | Activated Carbon Properties | Ref. |
|---|---|---|---|---|
| Bamboo powder | <0.5 mm | $CO_2$<br>Activation: 800 °C for 1–4 h | Activation time from 1–4 h;<br>$S_{BET}$: 1000, 1500, 2000 m²/g; | [106] |
| Coking coal | Special low ash coking coal; diameter grain 1.5 mm | KOH, $K_2CO_3$ (Mass ratio: 1:1), $KOH/ZnCl_2$, $ZnCl_2$ (20 wt%)<br>Impregnation: 3 h; temperature: 800 °C for 1 h | AC-KOH, AC-$K_2CO_3$, AC-KOH/$ZnCl_2$, AC-$ZnCl_2$;<br>$S_{BET}$: 1112, 1759, 1227, 1757, 1304 m²/g;<br>$V_{micro}$: 0.46, 0.67, 0.44, 0.66, 0.53 cm³/g; | [43] |
| Ma bamboo | - | $H_3PO_4$; 45 wt%;<br>Activation: 400 °C for 2 h | $S_{BET}$: 1692 ± 9 m²/g;<br>$V_{Total}$: 0.94 cm³/g | [121] |
| Walnut shells | 4-mesh size | KOH, $ZnCl_2$, $K_2CO_3$, $H_3PO_4$;<br>Carbonization: 400–800 °C for 1 h<br>Activation: 600–900 °C for 1 h | Activation: 500, 600, 700 °C;<br>$S_{BET}$: 678.30, 1239.92, 634.12 m²/g;<br>$V_{total}$: 0.463, 0.766, 0.246 cm³/g; | [116] |
| Walnut shells | V: 68.33%;<br>FC: 28.62%;<br>A: 3.05% | KOH; KOH/C = 4<br>Microwave heating: 700 W for 20 min, | $S_{BET}$: 3923 m²/g;<br>$V_{Total}$: 2.778 cm³/g;<br>$D_p$: 2.32 nm | [136] |
| Watermelon shell (WM), Walnut shells (WN) | Grain size: 2 mm | $H_3PO_4$ (40 wt%);<br>C/$H_3PO_4$ = 1:2;<br>Activation: 500–700 °C for 1–2 h | ACWM, ACWN;<br>$S_{BET}$: 710, 789 m²/g;<br>$V_{total}$: 0.263, 0.304 cm³/g | [108] |
| Brewing yeast | Size less than 0.15 mm | $Na_2CO_3$;<br>Brewing yeast: $Na_2CO_3$ = 4:1;<br>Temperature: 700 °C for 4 h | $S_{BET}$: 957.7 m²/g;<br>$V_{total}$: 0.81 cm³/g;<br>$D_p$: 4.9 nm | [5] |
| Sub-bituminous coal | - | $CO_2$: carbonization: 850 °C for 1 h and activation: 950 °C for 3–12 h;<br>KCl: activation 550 °C for 4 h;<br>$H_3PO_4$: activation 500 °C for 4 h; | AC-$CO_2$, AC-KCl, AC-$H_3PO_4$;<br>Highest $S_{BET}$: 1245, 1084, 1437 m²/g;<br>Highest $V_{total}$: 0.49, 0.46, 0.83 cm³/g | [107] |
| Cotton | - | KOH; KOH/char = 4;<br>Carbonization: 400 °C for 1 h; then mixed with KOH;<br>Activation: 800 °C for 0.5–1.5 h | Activation time: 0, 0.5, 1, 1.5 h;<br>$S_{BET}$: 6, 2823, 2600, 2263 m²/g;<br>$V_{total}$: 0.020, 1.56, 1.52, 1.22 cm³/g | [104] |
| Palm kernel shell (PKS) | <5 mm;<br>Lignin: 48 wt%;<br>cellulose: 30 wt%;<br>hemicellulose: 22 wt% | Microwave vacuum pyrolysis, then chemical activation with KOH and NaOH (CAC), or physical activation with water (PAC);<br>Carbonization and activation: microwave 700 W for 25 min; | PAC → CAC<br>$S_{BET}$ : 480 → 750 m²/g;<br>$V_{total}$ : 0.26 → 0.37 cm³/g;<br>$D_p$ : 4.4 → 4.8 nm | [127] |
| Anthracite coal | $M_{ad}$: 3.64%;<br>$A_d$: 3.70%;<br>$V_{daf}$: 13.79%;<br>$FC_{daf}$: 86.21% | NaOH, Steam;<br>Coal was mixed with NaOH solution;<br>Carbonization: 480 °C for 1 h;<br>Steam activation: 830 °C for 4 h | $S_{BET}$: 837.992 m²/g;<br>$V_{total}$: 0.372 cm³/g | [140] |
| Walnut shell | - | $H_3PO_4$; $H_3PO_4$/C = 1.25;<br>Soak: at 150 °C for 12 h;<br>Activation: 550 °C;<br>KOH; KOH/C = 1.25, 2.5, 3.5;<br>Activation: 700, 800, 900 °C | $S_{BET}$(KOH) > $S_{BET}$($H_3PO_4$), all higher than 1260 m²/g;<br>the highest $S_{BET}$ was 3143.47 m²/g; | [96] |
| Waste ion-exchange resin | V: 51.88%; A:10.83%;<br>FC: 37.29%; M: 2.7% | KOH; KOH: char = 0.5–2.5<br>Carbonization: 800 °C for 1 h;<br>Activation: 600–1000 °C for 1–3 h | Highest $S_{BET}$: 987.20 m²/g;<br>Pore size distribution: 0.9–4 nm | [141] |
| Carbon nanotubes | - | Air; Activation: 713 K for 40 min<br>$CO_2$; Activation: 1123 K for 4 h<br>KOH, ratio: 3:1; activation: 1123 K for 1 h. | AC-Air, AC-$CO_2$, AC-KOH;<br>$S_{BET}$: 270, 429, 785 m²/g;<br>$V_{micro}$: 0.06, 0.10, 0.17 cm³/g; | [95] |

Table 1. *Cont.*

| Raw Material | Raw Material Characteristics | Activation Agent and Main Parameters | Activated Carbon Properties | Ref. |
|---|---|---|---|---|
| Tea woody stem (TWS) | V: 76.59%; A: 1.69% | KOH; TWS/KOH = 1:2; Carbonization: 500 °C for 3 h. | $S_{BET}$: 789 $m^2/g$; $V_{total}$: 0.31 $cm^3/g$; $V_{micro}/V_{total}$ = 90.32% | [115] |
| Oil palm kernel shell | - | $ZnCl_2$; ratio: 0.55; Activation: 900 °C for 2.5 h | $S_{BET}$: 1548 $m^2/g$; $V_{total}$: 1.0 $cm^3/g$; Micropores: 58%, mesopores: 42% | [100] |
| Hazelnut shell | - | $H_2O$, $CO_2$, KOH, $H_3PO_4$, $ZnCl_2$, $K_2CO_3$, | $S_{BET}$: 1057, 968, 2031, 1333, 1643, 912 $m^2/g$; $V_{total}$: 0.577, 0.498, 0.930, 1.032, 1.058, 0.433 $cm^3/g$; | [102] |
| Crofton weed | 3–6 mm V: 75.96%; A: 1.86%; FC: 22.18% | KOH; Carbonization: 500 °C 2 h; KOH: char = 4; Microwave heating: 700 W 15 min | $S_{BET}$: 61 (char), 3918 (AC) $m^2/g$; $V_{total}$: 0.109 (char), 2.383 (AC) $cm^3/g$; $D_p$: 7.15(char), 2.43(AC) nm | [94] |
| Rise husk biomass | - | Carbonization: 500 °C 2 h; Heating rate: 7 °C/min | $S_{BET}$: 1.39 $m^2/g$; $V_{total}$: 0.016 cc/g; $D_p$: 3.52 nm | [41] |
| Crushed olive stone | 1.25–3.00 mm; C: 47.3%; H: 6.1%; O: 45.85%; N: 0.075%; A: 1.86% | Steam; Carbonization: 600 °C for 2 h; Activation: 850 °C for 8 h | $S_{BET}$: 737.17 $m^2/g$; $V_{total}$: 0.407 $cm^3/g$; $D_p$: 2.22 nm | [38] |
| Coal | - | Steam activation | $S_{BET}$: 909 $m^2/g$; $V_{micro}$: 0.332 $m^3/g$ | [72] |
| Olive stone | - | $H_3PO_4$ (50 wt%); Temperature 170 °C (30 min), then 380 °C (150 min); $N_2$ atmosphere | $S_{BET}$: 1174 $m^2/g$; V: 0.46 $cm^3/g$ Micropores: 94% total pore volume; Adsorption: 93% amoxicillin (25 mg/L) | [138] |
| Nanoporous carbide-derived carbon | - | Argon/water vapor mixture; Temperature 900 °C for 60–75 min | $V_{total}$ and $D_p$ increase with increasing temperature. | [133] |
| Oil palm shell | 0.5–1.5 mm V: 64.7%; FC: 26.4%; A: 2.2% | KOH (85% wt%); Raw: KOH = 1:2; Temperature 500–900 °C for 3 h | $S_{BET}$ increased linearly with increasing temperature to 800 °C (2301 $m^2/g$); at 900 °C, $S_{BET}$ decreased to 1707 $m^2/g$. | [119] |
| Apricot stones | 1–3 mm; Low sulfur (0.03%) and ash (0.81%) | Carbonization: 750–850 °C for 2–4 h; Activation with steam at 750–950 °C for 2–4 h; | 750 °C → 800 °C → 850 °C $S_{BET}$ : 95 → 110 → 140 $m^2/g$ | [44] |
| Shrimp shells | 0.149 mm; Abundant hierarchical structure and high graphitic degree | Carbonization: 400, 600, 800 °C for 2 h | $S_{BET}$: 3.3, 376, 594 $m^2/g$; $V_{total}$: 0.024, 0.38, 0.93 cc/g; $D_p$: 14.2, 2.0, 3.1 nm | [33] |
| Petroleum coke | 0.2–0.45 mm FC: 85.7%; M: 1.2%; V: 11.2% | KOH; KOH/coke = 1.0; Heating: 450 °C for 2 h; Then carbonization: 850 °C for 2 h under $N_2$ atmosphere. | $S_{BET}$ : < 30 → 1619 $m^2/g$; $V_{micro}$ : 0.02 → 0.55 $cm^3/g$; | [142] |
| Glucose | - | Hydrothermal carbonization: 160 °C for 4 h | $S_{BET}$: 5.9 $m^2/g$; $V_{total}$: 0.034 $cm^3/g$ | [143] |
| Sugar beets pulp | 65–80% polysaccharides, 40% cellulose, 30% hemicelluloses, 30% pectin | Physical activation; Temperature: 100–1000 °C for 2 h | $S_{BET}$: 880 $m^2/g$; $V_{total}$: 8.5 $cm^3/g$; $D_p$: 25 nm | [144] |
| Sludge | Ash: 50.60%; C: 25.88% | Pyrolysis: 300, 500, 700, 900 °C for 2 h; $N_2$ atmosphere | BC500, BC700, BC900; $S_{BET}$: 0.407, 39.1, 95.9 $m^2/g$; | [56] |

Table 1. *Cont.*

| Raw Material | Raw Material Characteristics | Activation Agent and Main Parameters | Activated Carbon Properties | Ref. |
|---|---|---|---|---|
| Sludge | - | (AC1) $ZnCl_2$ (3.0 mol/L); (AC2) $H_2SO_4$ (3.0 mol/L); (AC3) $ZnCl_2$:$H_2SO_4$ = 1:2; 10 g dry sludge and 10 mL activation agent for 24 h, and then dried at 105 °C; activation: 550 °C for 1 h | AC1, AC2, AC3; $S_{BET}$: 18.3, 51.9, 179.9 m$^2$/g; $V_{total}$: 0.032, 0.056, 0.164 cm$^3$/g; $D_p$: 6.953, 4.287, 3.633 nm | [74] |
| Apricot stone | - | KOH; KOH: samples = 3; Carbonization: 600 °C for 4 h; Activation: 750 °C for 4 h | $S_{BET}$: 1621.0 m$^2$/g; $V_{total}$: 1.07 cm$^3$/g; $D_p$: 1.321 nm | [145] |
| Olive stones (OS) | 1–2 mm | $H_3PO_4$; 60%, 70%, 80%; 50 g OS was impregnation with 200 mL $H_3PO_4$ at 85 °C for 4 h; Heating at 500 °C for 2 h | 60%$H_3PO_4$/AC, 70%$H_3PO_4$/AC, 80%$H_3PO_4$/AC; $S_{BET}$: 257, 779, 1218 m$^2$/g; $V_{total}$: 0.123, 0.35, 0.6 cm$^3$/g; $D_p$: 0.954, 1.0 1.1 nm | [128] |
| Pruning waste of apple trees | - | Carbonization: 400–500 °C; Activation: 400, 550, 70%0 °C for 2 h, under $N_2$ atmosphere | BC, BC400, BC550, BC700; $S_{BET}$: 42.2, 231.2, 424.6, 486.7 m$^2$/g; $V_{total}$: 0.032, 0.120, 0.198, 0.230 cm$^3$/g; $D_p$: 2.99, 2.06, 1.87, 1.89 nm | [146] |
| Mangosteen peels (MP) | - | $ZnCl_2$; $ZnCl_2$:MP = 4:1; Activation: 600 °C for 0.5 h | MP, AC; $S_{BET}$: 3.787, 1621.8 m$^2$/g; $V_{total}$: 0.002, 1.805 cm$^3$/g; $D_p$: 5.78, 4.43 nm | [147] |
| Cocoa shell | - | Inorganic components (20% lime + 40% $ZnCl_2$ + 40% $FeCl_3$); Ratio: 100 g:100 g; Dry at 100 °C for 6 h; Micro-wave heating: 360 W (80 s); 480 W (80 s); 600 W (80 s); 960 W (160 s); 1200 W (160 s) | AC, AC-1.0, AC-1.5, AC-2.0; $S_{BET}$: 2.6, 619, 562, 541 m$^2$/g; $V_{total}$: 0.003, 0.315, 0.286, 0.270 cm$^3$/g; $D_p$:1.7, 4.8, 5.2, 5.5 nm | [129] |
| Cherry stone | 1.6–2.0 mm | Physical activation, $CO_2$; Carbonization: 900 °C for 2 h; Activation: 850 °C for 2 h | $S_{BET}$: 604 m$^2$/g; $V_{total}$: 0.70 cm$^3$/g | [148] |
| Raw rice straw | 2–3 mm | Chemical activation: $(NH_4)_2HPO_4$ (30 wt%); Carbonization: 700 °C for 1 h, and AC-0 obtained; AC-1: AC-0 preoxidized 200 °C for 2 h and impregnation with $(NH_4)_2HPO_4$; AC-2:AC-0 impregnation with $(NH_4)_2HPO_4$; AC-3:AC-2 preoxidized under air atmosphere at 200 °C for 2 h; | AC-0, AC-1, AC-2, AC-3; $S_{BET}$: 351, 525, 902, 1154 m$^2$/g; $V_{total}$: 0.186, 0.291, 0.432, 0.670 cm$^3$/g; $D_p$: 1.91, 2.22, 1.92, 2.32 nm | [149] |
| Orange peel (OP) | M: 9.2%; V: 76.52%; A: 3.09%; FC: 20.39%. | $ZnCl_2$, $K_2CO_3$; activator agent: OP = 1:1; Mixture was dried at 110 °C; Activation: 400–1000 °C for 1 h; | $S_{BET}$: 9-1352 (AC-Z), 804-1215 (AC-K) m$^2$/g; Morphology: irregular and heterogeneous surface (AC-Z), honeycomb-like (AC-K); $K_2CO_3$ was more effective than $ZnCl_2$ | [110] |
| Barley straw | M: 9.0%; V: 77.2%; A: 5%; FC: 17.3%. | Physical activation: Steam and $CO_2$; Activation temperature: 500–600 °C | The maximum $S_{BET}$ and $V_{micro}$ of AC-$CO_2$ were 789 m$^2$/g and 0.3268 cm$^3$/g, while it was 552 m$^2$/g and 0.2304 cm$^3$/g for AC-$H_2O$ | [131] |

**Table 1.** *Cont.*

| Raw Material | Raw Material Characteristics | Activation Agent and Main Parameters | Activated Carbon Properties | Ref. |
|---|---|---|---|---|
| Coffee, tobacco, corn cobs, cherry stones | Ash: 16.0% (coffee), 9.0% (tobacco), 1.0% (corn cobs), 0.2% (cherry stones) | Carbonization: 700 °C; Physical activation: $CO_2$, 800 °C for 30 min; Chemical activation: KOH, 700 °C for 30 min | AC with the most developed surface area was obtained by KOH activation, irrespective of the precursor used. | [113] |
| Olive-waste cake | - | $H_3PO_4$ (60 wt%); Impregnation ratio: 1.75; Pyrolysis temperature: 450 °C for 2 h | $S_{BET}$: 793 $m^2/g$; $V_{total}$: 0.59 $cm^3/g$; $D_p$: 4.2 nm | [137] |
| Pine sawdust | Less than 1 mm; Water content: 11.6% | Hydrothermal-pressure preconditioning protocol (160 °C, 300–309 psi), then activated at 900 °C for 1 h | $S_{BET}$: 990 $m^2/g$; $V_{micro}$: 0.344 $cm^3/g$; | [130] |
| Olive stone | - | $H_2SO_4$ (10 wt%); Impregnation rate: 1:1; Carbonization: 550 °C for 1 h | $S_{BET}$: 83.72 $m^2/g$; Porosity value: 69.75%; Ash: 4% | [150] |
| Sawdust of poplar | - | $ZnCl_2$, KOH; Pyrolysis: 300–600 °C for 30–120 min | AC-$ZnCl_2$, AC-KOH; $S_{BET}$: 2430.8, 1506.2 $m^2/g$; $V_{micro}$: 062, 0.32 cc/g; $D_p$: 1.65, 1.92 nm | [118] |
| Rice husk | - | $ZnCl_2$; Carbonization: 400 °C for 4 h; Activation: 750 °C for 1 h | $S_{BET}$: 320.16 $m^2/g$; $V_{total}$: 0.15 $cm^3/g$ | [85] |

$S_{BET}$: Specific surface area (BET method). $V_{micro}$: Micropore volume. $V_{total}$: Total pore volume. $D_p$: Average pore diameter. M: Moisture content (wt.%). V: Volatile matter (wt.%). FC: Fixed carbon (wt.%). A: Ash (wt.%).

Physicochemical Activation

Combining physical activation and chemical activation, the physicochemical activation method is obtained; that is, chemical reagents and materials are mixed to modify, and then the physical activation method is used for activation, which can fully utilize the advantages of both methods and flexibly adjust the pore structure of activated carbon [22]. The combination of chemical and physical activation can prepare highly microporous activated carbon. Moreno-Piraján et al. [107] prepared activated carbon with a large surface area and narrow micropore size distribution from coal by physicochemical activation.

Activation Time

To the best of our knowledge, activation time directly affects the burn-off degree, and there is a linear relationship between the activation time and burn-off degree [107,116,121]. Moreno-Piraján et al. [107] found that the increase in activation time will lead to an increase in the number of micropores, while the higher degree of burn-off will lead to the generation of micropore structure, the micropores will expand to the mesopores, and the total pore volume will increase. This result was in accordance with other studies [115,133]. Tsubota et al. [106] found that when the activation time increased from 2 h to 3 h, both the micropore volume and the total volume increased, and the specific surface area increased. Some studies have also found that when the activation time increased from 0 to 90 min, there is a highly significant effect on the specific surface area, but little effect on the micro-morphology. Moreover, when the activation time is 60 min, the obtained activated carbon has the largest specific surface area and pore volume [104]. Liu et al. [121] prepared activated carbon from Ma bamboo at different activation times of 30, 60, 90, and 120 min, and discovered that an increased activation time is suitable for pore structure development, and the 120 min activation time is the best condition.

Activation Temperature

The effect of activation temperature on the activation process is mainly reflected in pore structure and specific surface area [100,119,121,123,131]. When the activation temperature increases, the reaction rate increases. Low activation temperature leads to high micropore content, while high activation temperature will convert micropores to mesopores or macropores [116,119]. This result was in accordance with the study by Liu et al. [121] who found that when the activation temperature was higher than 350–400 °C, the specific surface area and pore volumes of the activated carbon increased, while when the activation temperature was higher than 450–500 °C, the specific surface area and pore volumes of the activated carbon decreased. Rashidi et al. [96] concluded that when the activation temperature was 973 K, it was more suitable for preparing activated carbon with a developed pore structure and high micro–mesopore volume. Käärik et al. [133] found that the total pore volume and average pore size increase with the increase in activation temperature (500, 600, 700, 800, 900, 1000 °C). Consequently, it is necessary to set the appropriate activation temperature based on the requirements.

### 3.2.4. Additives

With the metal and/or its salts added to the raw material, the carbonization path or activation process may be changed, and the pore structure of activated carbon will be affected as well. Representative additives include iron series, calcium series, and potassium hydroxide [105,151]. In the process of preparing activated carbon with phosphoric acid activation, Tsubota et al. [106] used guanidine phosphate as an additive, and found that the increase in the amount of guanidine phosphate caused an increase in the specific surface area of activated carbon, although the guanidine phosphate is not the main reason. Tian et al. [140] used coal tar pitch as an additive to prepare activated carbon from anthracite coal, and concluded that the prepared activated carbon had a large surface area, abundant micropore structure, and high mechanical strength. Gong et al. [105] regulated the pore structure of coal-based activated carbon by potassium-catalyzed steam activation, and found that with the increase in KOH content, the average pore size and mesopores rate increased. Moreover, KOH played two main roles in the preparation of activated carbon: providing a reaction path and catalyzing the steam reaction process. Coal tar pitch as an additive is beneficial to improving the textural properties of the precursor. In the process of preparing manganese oxide catalyst, the addition of potassium, calcium, and magnesium metal enhanced the catalytic activity of the $Mn_3O_4$ catalyst, and the authors explained that the three additives caused a defect-oxide or a hydroxyl-like group, which is beneficial to improving the catalytic activity [152].

The dispersion of additives is of great significance to the pore structure regulation of activated carbon; the dispersion and particle size of active metals were affected by the loading methods and the property of the support (for example pore structure and specific surface). Methods to influence the dispersion include the mixing method, dipping method, and ion exchange method. The addition of additives in the raw materials by mixing and impregnation may inhibit the activation reaction to a certain extent, and the effect of pore structure regulation was not significant [153].

### 3.2.5. Modification

Besides direct preparation, modification is also a simple and low-cost method to prepare activated carbon, so it has been widely used [21,35,41,43,96,126,154]. Although the modification is mainly to change the surface properties of activated carbon, such as the introduction of acid or alkaline functional groups, as well as the change in oxygen functional groups on the surface of activated carbon, it can also adjust the pore structure of activated carbon [27,35,41,62,69,87,92,122,155]. Zhang et al. [41] found that all modified activated carbons exhibited rough and uneven surfaces; this is because the interactions took place between chemical reagents and the carbon matrix, so modification will obviously affect the pore structure of activated carbon. Vega et al. [32] modified three commercial

activated carbons to change the chemical structure and surface properties of activated carbons, which was conducive to the full contact between $H_2O_2$ and activated carbons and will therefore promote the decomposition of $H_2O_2$ to produce hydroxyl radicals, and greatly improve the degradation efficiency of the target pollutant.

There are various modification methods: heat treatment under different atmospheres, ozone, $H_2O_2$, $HNO_3$, nitrogenation, sulfuration treatment, etc. [45,148,156]. Acid, alkali, and oxidation were the typical treatment methods, and the common chemical reagents were $KMnO_4$, $HNO_3$, and NaOH [41,69,92,157]. Table 2 summarizes the modification effect on the pore structure of activated carbon. Because different raw materials always have different physical and chemical properties, it is necessary to select appropriate modifiers and modification methods [148,157]. Zhang et al. [41] modified the biochar derived from rice husk with acid (sulfuric acid and phosphoric acid), alkali (sodium hydroxide and sodium bicarbonate), and hydrogen peroxide, and the results showed that the alkali-modified activated carbon had the largest specific surface area and the best adsorption ability; however, the specific surface and pore volume of acid-modified activated carbon did not change significantly. Fang et al. [35] found that activated carbon modified by peracetic acid solution can improve the specific surface area and the catalytic activity (the benzene removal efficiency remained 100%), and the ozone utilization for benzene oxidation was also enhanced. Yang et al. [154] confirmed that after nitric acid oxidation by heat treatment in a nitrogen atmosphere, the surface area of activated carbon will increase. Gopinath et al. [27] stated that for activated carbon fiber, all the surface treatments will reduce the surface area and pore volume except thermal treatment. Li et al. [157] modified granular activated carbon for improving adsorption ability by ammonia, sodium hydroxide, nitric acid, sulfuric acid, and phosphoric acid, and the result showed that the surface area increased slightly and total pore volume decreased when AC was treated by acid; meanwhile, total pore volume increased when AC was treated by alkalis, and in all cases, the total micropore volume of alkali-modified AC was slightly higher than that of acid-modified AC. Stavropoulos et al. [155] studied the effect of different modification methods on activated carbon and concluded that nitric acid modification significantly reduced the pore volume of activated carbon, while thermal partial oxidation by oxygen increased the pore volume. Ozone will modify the activated carbon during the reaction, resulting in changes in the surface functional groups and pore sizes of the activated carbon [46,64]. Abdedayem et al. [46] proposed that after ozone treatment, the specific surface area of activated carbon remained unchanged, but the pore volume increased slightly, and the mesoporous volume decreased a little. It should be noted that the decrease in surface area after modification did not mean micropore collapse, but perhaps the functional group was occupied [156].

However, although the modification method is widely used, it mainly affects the surface properties of activated carbon (such as $pH_{pzc}$ and surface groups), and its effect on the pore structure of activated carbon is relatively small [21,92].

**Table 2.** Effect of modification on the properties of activated carbon.

| Activated Carbon (AC) | Modifier | Main Parameters | Performance | Ref. |
|---|---|---|---|---|
| Commercial AC: Norit GAC 1240 PLUS | $HNO_3$, 6 mol/L | Boiling temperature; 3 h | $S_{BET}$ : $909 \rightarrow 827$ $m^2/g$; $V_{micro}$ : $0.332 \rightarrow 0.304$ $cm^3/g$; the ability of adsorbing and catalyzing oxalic acid decreased | [69] |
| Commercial coal-based AC | $C_2H_4O_3$, $\geq$12%; 50 mL | 100 °C for 3 h | $S_{BET}$ : $402.6 \rightarrow 1051$ $m^2/g$; | [35] |

**Table 2.** *Cont.*

| Activated Carbon (AC) | Modifier | Main Parameters | Performance | Ref. |
|---|---|---|---|---|
| Rice husk biomass biochar | $H_2SO_4$, $H_3PO_4$, NaOH, $NaHCO_3$, $H_2O_2$ | Microwave-assisted chemical modification; Microwave: 600 W for 2 h | Rough and uneven surface; Alkali modified AC had the largest $S_{BET}$ (119, 102 m$^2$/g), $D_P$ increased; acid modified AC had no significant changes of $S_{BET}$ and $D_P$ | [41] |
| Activated carbon fiber (ACF) | HCl 10%; $HNO_3$, 3 mol/L | ACF-0: ACF + 10% HCl, adjust pH to 7.0, dry at 100 °C for 24 h; ACF-N: ACF-0 + 250 mL $HNO_3$; ACF-H: ACF-0 + 900 °C; ACF-NH: ACF-N + 900 °C | ACF-0, ACF-N, ACF-H, ACF-NH; $S_{BET}$: 1148, 216, 1058, 816 m$^2$/g; $V_{micro}$: 0.108, 0.052, 0.062, 0.098 cm$^3$/g | [154] |
| Coconut shell commercial AC; 60–80 mesh | $HNO_3$ (10 mol/L); $H_2SO_4$ (9.0 mol/L); $H_3PO_4$ (7.3 mol/L); NaOH (10 mol/L); $NH_3H_2O$ (6.6 mol/L) | AC was soaked in each solution at 70 °C for 2 h; Then place on a rotation at 35 °C for 24 h | AC, AC/$NH_3H_2O$, AC/NaOH, AC/$H_2SO_4$, AC/$HNO_3$, AC/$H_3PO_4$; $S_{BET}$: 731, 868, 846, 528, 608, 441 m$^2$/g; $V_{total}$: 0.168, 0.176, 0.178, 0.170, 0.156, 0.150 cm$^3$/g; $D_P$: 2.346, 2.391, 2.372, 2.514, 2.358, 2.442 nm | [157] |
| Wood-derived AC powder | Heat and oxygen | Initial heat treatment (ACN): AC was calcined at 1000 °C for 5 h under $N_2$ atmosphere; Oxidation treatment (ACNO): CAN was calcined at 400 °C for 5 h; Second heat treatment (ACNON): ACNO was calcined at 950 °C for 5 h under $N_2$ atmosphere | AC, ACN, ACNO, ACNON; $S_{BET}$: 1180, 1170, 1400, 1400 m$^2$/g; $V_{micro}$: 0.54, 0.56, 0.62, 0.63 cm$^3$/g. | [62] |
| AC carbo Tech D45/2; Less than 180 μm | $HNO_3$; $O_2$; urea | (1) AC-N: liquid phase oxidation by $HNO_3$; (2) AC-O: thermal partial oxidation by $O_2$; (3) AC-UI: thermal treatment of urea pre -impregnation samples; (4) AC-UH: thermal treatment under a urea saturated helium flow | AC, AC-N, AC-O, AC-UI, AC-UH; $S_{BET}$: 1003, 260–725, 917–1092, 653–712, 468–563 m$^2$/g; $V_{total}$: 0.374, 0.116–0.263, 0.369–0.526, 0.294–0.334, 0.207–0.227 cm$^3$/g. | [155] |
| AC; 100–400 mesh, powder | $HNO_3$; $NH_3H_2O$ | AC-$NO_2$: 1 g AC with 30 mL fuming nitric acid, stirring for 5 h, then 19 h, drying at 378 K for 12 h; AC-$NH_2$: 1 g AC-$NO_2$ with 10 mL strong aqua ammonia, stirring at 293 K for 24 h, then drying at 378 K for 12 h | AC, AC-$NO_2$, AC-$NH_2$; $S_{BET}$: 1128, 232, 1023 m$^2$/g; $V_{micro}$: 0.269, 0.055, 0.263 cm$^3$/g | [156] |
| Cherry stone AC | $O_2$, $O_3$, $HNO_3$, $H_2O_2$ | $O_2$, 300 °C for 24 h; $O_3$ (1.25 wt%), 100 °C for 1 h; $HNO_3$ (5 mol/L), 90–95 °C for 10 h; $H_2O_2$ (5 mol/L), 20 °C for 10 h | AC, AC-$O_2$, AC-$O_3$, AC-$HNO_3$, AC-$H_2O_2$; $S_{BET}$: 604, 469, 603, 399, 591 m$^2$/g; $V_{total}$: 0.70, 0.61, 0.71, 0.62, 0.69 cm$^3$/g | [148] |

**Table 2.** *Cont.*

| Activated Carbon (AC) | Modifier | Main Parameters | Performance | Ref. |
|---|---|---|---|---|
| AC: commercial AC; PAC: petroleum coke AC | Ammonia; nitric acid and sulfuric acid | Ammonia modification: 700 °C for 3 h; AC-NH$_3$, PAC-NH$_3$; nitric acid and sulfuric modification: 105 °C for 24 h; AC-NH$_2$, PAC-NH$_2$ | AC, AC-NH$_2$, AC-NH$_3$, PAC, PAC-NH$_2$, PAC-NH$_3$; S$_{BET}$: 1399, 955, 1266, 1443, 967, 1024 m$^2$/g; V$_{total}$: 0.70, 0.5, 0.6, 0.8, 0.5 0.5 cm$^3$/g; D$_p$: 2.0, 2.0, 2.0, 2.1, 2.1, 2.1 nm | [158] |
| Municipal sewage sludge biochars (MSAC) | NaOH (2 mol/L); HNO$_3$ (14 mol/L) | First modified with NaOH at 90 °C for 2 h, then modified with HNO$_3$ at 10–15 °C for 2 h, then SNMS was obtained. | MS-400, MS-600, MS-800, SNMS-400, SNMS-600, SNMS-800; S$_{BET}$: 39.08, 43.58, 67.39, 37.59, 153.72, 202.52 m$^2$/g; V$_{total}$: 0.0676, 0.0742, 0.0885, 0.0932, 0.186, 0.2563 cc/g; D$_p$: 3.73, 3.72, 3.74, 3.31, 3.89, 3.86 nm | [83] |

S$_{BET}$: Specific surface area (BET method). V$_{micro}$: Micropore volume. V$_{total}$: Total pore volume. D$_p$: Average pore diameter. M: Moisture content (wt.%). V: Volatile matter (wt.%). FC: Fixed carbon (wt.%). A: Ash (wt.%).

### 3.2.6. Load

In heterogeneous catalytic ozonation, activated carbon is often used as a carrier supporting transition metals or their oxides to prepare a catalyst, such as Mn, Fe, Cu, Co, Ce, Zn, Ti, etc. [159]. The transition metals supported on activated carbon were usually prepared by impregnation, the deposition–precipitation method, ion exchange, and the sol–gel method [36,46,54,98]. The activated carbon catalyst loaded with metal oxides has more active sites and higher catalytic activities [6,18,84,160]. Moreover, the pore size and specific surface area of activated carbon are affected by the metal activity group [23,34]. Table 3 summarizes the loading effect on the properties or performance of activated carbon catalysts. Huang et al. [5] prepared catalysts by loading metals such as cobalt, copper, iron, and manganese on activated carbon, and they found that compared to activated carbon, the activated carbon catalysts showed a significantly smaller BET surface area and pore volume as well as slightly smaller average pore size, and there was an obvious decrease in adsorption capability. The possible reason is that part of the metal atoms entered into the pores of support and blocked pores, or the metal atoms stayed at the surface of activated carbon and reduced its surface area. Compared to the activated carbon, Tang et al. [34] proposed that the Fe-Mn activated carbon would reduce the surface area and pore volume; it may be the Fe and Mn oxides enter the pores of the activated carbon and block some of the pores, but the adsorption ability of activated carbon catalyst was almost not changed. Srilatha et al. [23] found that after loading with nickel, the BET surface and pore volume of activated carbon will lose about 50%, and the catalytic activity will decrease. However, Li et al. [6] stated that the surface area of Fe-activated carbon catalyst did not decrease, it may be that Fe$_2$O$_3$ dispersed well on the activated carbon and did not block the partial pores of activated carbon. Akhtar et al. [161] proposed that the impregnation of CeO$_2$/Fe$_2$O$_3$ into activated carbon will decrease the pore volume and convert some mesopores into micropores, and the adsorption of metal oxides will form new micropores, resulting in a 4.7% decrease in mesopore volume and 4.8% increase in micropore volume. Dabuth et al. [2] confirmed that the loading of TiO$_2$ by the sol–gel method increased the micropore and mesopore volume of commercial coal-based and coconut shell-based activated carbon catalysts. Huang et al. [88] studied the effect of iron loading on the characteristics of activated carbon and confirmed that the surface area, pore volume, and pore diameter all decreased, and the catalyst with 15% iron content achieved the highest catalytic effect. The authors also proposed that although metal loading on activated carbon will increase catalytic activity, increasing iron loading (the iron content increase from 15% to 30%) will decrease the DBP (dibutyl phthalate) removal rate, because overloaded iron species might enhance steric hindrance and reduce the surface area of the catalyst.

In short, the deposition of metal active components on the surface of activated carbon will block part of the pores, which will reduce the specific surface area and pore size of activated carbon, the degree of reduction depending on loading concentration and the dispersion of active component [46,72,91,127,161–164]. However, it should be noted that the catalytic activity may not decline, because the supported active components have a catalytic effect, which makes up for the decline of catalytic activity caused by the change in activated carbon porosity [30,163]. Huang et al. [74] confirmed that excessive loading (45% Mn load on sludge-activated carbon) would lead to serious plugging of pores and rapid reduction of the specific surface area of activated carbon, and decline of catalytic activity. Therefore, the appropriate load should be determined according to the actual demand and should not be too large [7,88,159].

**Table 3.** Effect of loading on the properties and performance of activated carbon catalyst.

| Activated Carbon | Active Compounds | Loading Method and main Parameters | Properties or Performance of Catalyst | Ref. |
|---|---|---|---|---|
| Crushed olive stones AC | Co; 5 wt% | Impregnation; Immersed time: 24 h; Calcination: 550 °C for 2 h | $S_{BET}$: 734.17 → 721.92 m$^2$/g; $V_{micro}$: 0.34 → 0.40 cm$^3$/g; $V_{total}$: 0.407 → 0.457 cm$^3$/g; | [46] |
| Commercial AC | Ce-O, 45 wt% | Precipitation; Calcination: 450 °C for 3 h | $S_{BET}$: 909 → 583 m$^2$/g | [72] |
| AC; 0.4 mm | MnO$_x$ | Impregnation; Calcination: 450 °C for 3 h, N$_2$ 50 cm$^3$/min | $S_{BET}$: 854.7 → 779.6 m$^2$/g; $V_{total}$: 0.410 → 0.361 cm$^3$/g; $D_p$ increased; microporous. | [30] |
| Brewing yeast | Co, Cu, Fe, Mn; 400 mL 2000 mg/L | Impregnation; Dry: 60 °C | AC, Co/AC, Cu/AC, Fe/AC, Mn/AC; $S_{BET}$: 957.7, 789.7, 485.3, 486.1, 529.8 m$^2$/g; $V_{total}$: 0.81, 0.67, 0.45, 0.75, 0.47 cm$^3$/g; $D_p$: 4.9, 3.4, 3.7, 3.8, 3.5 nm | [5] |
| AC; 0.6–2.0 mm | Fe | Dipping method; Ratio: Fe: AC = 1%; Calcination: 450 °C for 2 h, | $S_{BET}$: 536 → 586 m$^2$/g; $V_{total}$: 0.287 → 0.305 cm$^3$/g; $D_p$: unchanged; | [6] |
| Sludge-base AC | Fe; 2.3 wt%, 4.3 wt%, 9.5 wt% | Co-precipitation method | AC, 2.3%Fe/AC, 4.3%Fe/AC, 9.5%Fe/AC, $S_{BET}$: 941.0, 940.9, 936.7, 880.6 m$^2$/g; $V_{micro}$: 0.41, 0.4, 0.4, 0.35 cm$^3$/g; $D_p$: 22.23, 22.25, 22.59, 24.44 nm | [84] |
| Commercial AC; 0.45 mm | Ce; 0.3 wt%, 1 wt% | Impregnation; Heat: 250 °C for 2 h | AC, 0.3%Ce/AC, 1%Ce/AC; $S_{BET}$: 893, 881, 804 m$^2$/g; $V_{total}$: 0.353, 0.335, 0.322 cm$^3$/g; $D_p$: 2.177, 2.218, 1.803 nm | [7] |
| Granular AC | Nano size Fe$_3$O$_4$ | Impregnation | $S_{BET}$: 375.8 → 341.2 m$^2$/g; $V_{micro}$: 0.195 → 0.172 cm$^3$/g; $D_p$: 5.932 → 5.166 nm | [92] |
| Coal-based AC, 2.0 mm | Fe-Mn bi-metallic oxide | Impregnation-desiccation method; Calcination: 773 K for 3 h | $S_{BET}$: 873 → 785 m$^2$/g; $V_{micro}$: 0.351 → 0.345 cm$^3$/g; $V_{total}$: 0.485 → 0.442 cm$^3$/g | [34] |
| Crushed olive stone AC | Co; 5 wt% | Wetness impregnation method; Heat: 550 °C for 2 h | $S_{BET}$: 734.17 → 721.92 m$^2$/g; $V_{micro}$: 0.34 → 0.35 cm$^3$/g; $V_{total}$: 0.407 → 0.423 cm$^3$/g; $D_p$: 2.22 → 2.36 nm | [38] |
| Coconut shell granular AC, 40–60 mesh | CuO; 1%, 3%, 5%, 7%, wt% | Impregnation method; Calcination: 350 °C for 3 h | Pretreated-AC, AC, CuO-1%/AC, CuO-3%/AC, CuO-5%/AC, CuO-7%/AC; $S_{BET}$: 904.33, 1280.00, 1430.46, 1244.65, 896.64, 754.23 m$^2$/g; $D_p$: 2.00, 2.08, 2.02, 2.84, 2.43, 2.03 nm | [36] |

**Table 3.** *Cont.*

| Activated Carbon | Active Compounds | Loading Method and main Parameters | Properties or Performance of Catalyst | Ref. |
|---|---|---|---|---|
| Commercial AC | Fe | Impregnation method; 55 °C for 24 h | $S_{BET}$ : 1062.00 → 707.50 m$^2$/g; $V_{total}$ : 0.66 → 0.44 cm$^3$/g; | [162] |
| Granular AC | $MnO_2$-$Co_3O_4$ | Impregnation-precipitation method; Calcination 500 °C for 1 h | Compared with AC, $MnO_2$-$Co_3O_4$/AC was more efficient at COD removal | [49] |
| Coconut hull AC, 0.45 mm | Ce | Dipping method; Dipping: 30 °C for 2 h; Heating: 450 °C for 2 h | $S_{BET}$ : 632.16 → 522.98 m$^2$/g; $V_{total}$ : 0.450 → 0.272 cm$^3$/g; $D_p$ : 2.845 → 2.079 nm | [163] |
| Palm shell-based AC | Fe, Ce | Impregnation method; Calcination: 600 °C for 3 h | $S_{BET}$ : 1607.6 → 1536.3 m$^2$/g; $V_{total}$ : 0.5904 → 0.5916 cm$^3$/g; $V_{micro}$ : 0.37 → 0.39 cm$^3$/g | [161] |
| AC | Fe | Incipient wetness impregnation; | $S_{BET}$: 195.0 m$^2$/g; $V_{total}$: 0.165 cm$^3$/g; $D_p$: 3.374 nm | [87] |
| Coal-based AC (coAC); coconut Shell-based AC (ccAC) | $TiO_2$; | Sol–gel method; Dry at 100 °C for 4 h in a hot-air oven; then calcined at 400 °C for 4 h in a furnace. | ccAC, coAC, $TiO_2$-ccAC, $TiO_2$-coAC; $S_{BET}$: 633.19, 842.45, 805.22, 809.30 m$^2$/g; $V_{micro}$: 0.179, 0.227, 0.209, 0.203 cm$^3$/g; $D_p$: 2.164, 2.177, 2.150, 2.318 nm | [2] |
| Carbon nanotube (CNT) | $CeO_2$; 4.8%, 7.4%, 9.0%, 14.1% wt% | Hydrothermal method; Calcination: 300 °C for 3 h | $CeO_2$ loading 4.8%, 7.4%, 9.0%, 14.1% $S_{BET}$: 271.3, 244.3, 233.9, 220.3 m$^2$/g; $V_{total}$: 0.85, 0.65, 0.65, 0.67 cm$^3$/g; $D_p$: 1.26, 1.23, 1.18, 1.21 nm | [165] |
| Powder AC | Fe, Mn | Pechini sol–gel method; Calcination: 400 °C for 2 h | AC powder, Fe-Mn/AC powder, Fe-Mn/AC pellet; $S_{BET}$: 847.57, 737.31, 682.14 m$^2$/g; $V_{total}$: 0.208, 0.177, 0.143 cm$^3$/g; $D_p$: 3.827, 3.726, 3.021 nm | [54] |
| Sludge-based AC (SBAC) | Fe (15.23%), Mn (7.51%) | Wet impregnation; Calcination: 600 °C for 3 h | SBAC, $MnO_x$/AC, $FeO_x$/AC; $S_{BET}$: 398.6, 327.5, 339.1 m$^2$/g; $V_{micro}$: 0.141, 0.122, 0.127 cm$^3$/g $D_p$: 3.725, 3.318, 3.371 nm. | [91] |
| Coal-based AC; <37 μm | Fe; 1%, 5%, 15%, 30% | Dipping method; Heating at 40 °C for 48 h; Dry: 100 °C for 12 h | AC, 15% Fe-AC, 30% Fe-AC; $S_{BET}$: 964, 909, 777 m$^2$/g; $V_{total}$: 0.436, 0.434, 0.371 cm$^3$/g; $D_p$: 1.41, 1.35, 1.18 nm | [88] |
| Carbon spheres (CS) | Co, Ni | Dipping method; Drying at 60 °C | CS, Co@CS, Ni@CS, Co/Ni@CS; $S_{BET}$: 5.9, 12.2, 7.1, 20.6 m$^2$/g; $V_{total}$: 0.034, 0.073, 0.024, 0.126 cm$^3$/g; | [143] |
| Sludge AC (SAC) | Mn; 5%, 15%, 30%, 45% | Wet impregnation; 5 g SAC were dipped into 40 mL $KMnO_4$ solution, drying at 105 °C, calcined at 550 °C for 1 h. | 15%Mn/SAC, 30%Mn/SAC, 45%Mn/SAC; $S_{BET}$: 11.7, 10.9, 3.7 m$^2$/g; $V_{total}$: 0.025, 0.023, 0.009 cm$^3$/g; $D_p$: 8.501, 8.471, 9.845 nm | [74] |
| Coal-based GAC | Mn; 0.44%, 1.1%, 5.5%, 11% | In situ reduction of $KMnO_4$ with AC. | 0.44%Mn: porous lichen-like structure; 1.1%Mn: lichen-like structure, $MnO_x$ layer became denser; 5.5%Mn: compact nanosphere; 11%Mn: $MnO_x$ nanobelts were stacked into nanospheres. | [166] |

$S_{BET}$: Specific surface area (BET method). $V_{micro}$: Micropore volume. $V_{total}$: Total pore volume. $D_p$: Average pore diameter. M: Moisture content (wt.%). V: Volatile matter (wt.%). FC: Fixed carbon (wt.%). A: Ash (wt.%).

### 3.3. Influence of Pore Size and Distribution on Catalytic Ozonation

For activated carbon catalysts, the influence of pore size and its distribution on catalytic ozonation is mainly reflected in the adsorption, catalytic efficiency, and diffusion [39,53,156].

As described in chapter 2, activated carbon can greatly improve catalytic efficiency, mainly because it can promote the generation of free radicals, provides a large number of catalytic active sites, and has powerful adsorption ability. The most important factors affecting adsorption are pore structure and specific surface area, and catalytic active sites and the generation of free radicals are also related to the specific surface area of activated carbon. Many studies have shown that in catalytic ozonation, adsorption contributes a lot and even plays a major role in the degradation of wastewater. This is because ozone molecules and/or target pollutants need to be adsorbed on the surface of activated carbon to form hydroxyl radicals, which will oxidate pollutants with high efficiency [46,52]. Catalytic efficiency mainly depends on the quantity and production rate of oxygen free radicals (mainly hydroxyl radicals), because compared with adsorption and direct oxidation, hydroxyl radicals can degrade organic pollutants more efficiently and non-selectively. However, many factors affect the generation of free radicals, mainly including operating factors (such as solution pH, initial ozone concentration, reaction temperature, etc.) and the catalyst properties (pore structure, specific surface area, surface functional groups, active components, etc.) [51].

Within a certain range, the increase in average pore diameter, pore volume, and specific surface area of the catalyst not only increases the adsorption capacity but also greatly promotes the catalytic efficiency, which has been proved by various studies [1,52]. Fang et al. [35] reported that the adsorption ability of carriers is very important to the degradation of benzene in the catalytic ozonation process. When the coal-based activated carbon was modified with peracetic acid solution, authors found that the specific surface area of activated carbon rose from 402.6 $m^2$/g to 1051.0 $m^2$/g, and the modified catalyst for benzene processing efficiency significantly increased. Although modification affects the catalytic efficiency by affecting oxygen-containing functional groups on the surface of activated carbon, the increase in the specific surface area is also an important factor affecting the catalytic efficiency [79]. Moreover, Messele et al. [53] proposed that there was a correlation between the rate constant of 1,3-Adamantanedicarboxylic acid (ADA) by catalytic ozonation and the total pore volume of the tested activated carbon. However, there are few studies on the quantitative relationship between pore structure characteristics (average pore size, pore volume, and specific surface area) of activated carbon and removal efficiency.

The specific surface area can be divided into the total specific surface area, internal specific surface area, and external specific surface area; however, the contribution of internal and external specific surface area to catalytic efficiency of the reaction is often different in different circumstances [79]. Chen et al. [39] found that besides the total specific surface area, the higher the proportion of the external specific surface area, the higher the catalytic efficiency, because the external specific surface of the catalyst is the main site for adsorption and reaction, not the internal specific surface. In addition, Biernacki et al. [79] confirmed that the external specific surface is the main site for the decomposition of ozone molecules into hydroxyl radicals, and they further subdivided the specific surface area of activated carbon into the total specific surface area, geometric external specific surface area, external specific surface area, and microporous specific surface area. It was also confirmed that when the particle size of activated carbon decreased, the catalytic efficiency and ozone decomposition rate increased significantly in the first few minutes [79]. Fijolek et al. [28] also proposed that the outer surface of the activated carbon rather than the inner surface determines the generation of hydroxyl radicals. Although the inner surface also adsorbs part of ozone molecules and organics, the effect is minimal. How to improve the proportion of external surface area in total surface area is an effective method to improve catalytic efficiency.

Reasonable pore size distribution is also an important factor. Reasonable pore size distribution (especially the distribution of mesopores in a certain proportion) and the increase in useful specific surface area will lead to an increase in the number of active sites and the dispersion uniformity of active components on the catalyst surface. As a result, the number and the generation rate of free radicals will increase, which will improve the

catalytic efficiency of the target pollutant. Chen et al. [39] found that when the particle size of granular activated carbon increases from 10 mesh to 800 mesh, the specific surface area of activated carbon and the TOC removal rate of oxalic acid increased significantly. With the increase in mesopore, macropore, and specific surface area, the adsorption capacity and catalytic efficiency of the catalyst increased. In addition, the contribution of surface reaction caused by the change of particle size to the TOC removal effect also increased from 61% of the activated carbon before treatment to 96% of the activated carbon after treatment. This also proved that although the specific surface area of activated carbon is mainly contributed by micropores, a certain proportion of mesopores and macropores are also necessary. This is because, although the inner surface of the catalyst has adsorption capacity and contributes a certain amount of specific surface area, the amount of adsorbents entering the inner surface through micropores is relatively small due to the size of the absorbents, resulting in a limited catalytic efficiency [39]. Under this premise, the surface utilization efficiency can be improved by increasing the proportion of mesopores and macropores as channels. However, there is no research on the specific proportion or scale range of the appropriate macropores, mesopores, and micropores.

At present, the effect of the pore structure of activated carbon on catalytic ozonation is relatively shallow, mainly focusing on changes in the pore structure of activated carbon (especially the increase in the specific surface area) that will increase or decrease catalytic activity, but there is no quantitative relationship between pore structure parameters and catalytic efficiency. Catalytic efficiency can be improved by increasing the external surface area and the proportion of mesopores.

## 4. Influence of Pore Structure on Mass Transfer

The mass transfer of ozone molecules is an important factor affecting catalytic efficiency, because it will affect the generation rate of free radicals. Beltrán et al. [167] believed that the mass transfer process of ozone molecules from aqueous solution to the surface of activated carbon is affected by the decomposition of ozone molecules on the surface of activated carbon. They also pointed out that the chemical reaction rate and mass transfer resistances are of similar order of magnitude under these conditions. In another paper, they stressed that in addition to gas–liquid mass transfer resistance, there are external liquid–solid and internal diffusion mass transfer resistance. Powdered activated carbon will eliminate the external liquid–solid and internal pore mass transfer resistances, but it is highly dependent on gas–liquid mass transfer resistance [52]. Many studies have also proved that the introduction of activated carbon can improve the liquid–solid contact efficiency and enhance mass transfer [30,79].

Researchers have established several relevant models to study the mass transfer process. In the process of catalytic ozonation of indoles, Jiang et al. [93] considered that the mass transfer model of ozone molecules could be regarded as a gas–liquid double-membrane model, and the mass transfer process could be divided into three stages: ozone molecules transfer from gas phase body to the gas phase interface, then from the gas phase interface to the liquid phase interface, and finally, from the liquid phase interface to the liquid phase body, that is gas ozone molecules–gas phase interface–liquid phase interface-solution body. In this process, the mass transfer rate from the gas phase interface to the liquid phase interface is relatively slow, which is a restrictive step. The mass transfer equation is established according to the equilibrium between the mass transfer rate and the reaction rate. Based on the double-membrane theory, some researchers have proposed that the mass transfer rate of ozone from the gas phase to the liquid phase was the function of the volumetric mass transfer coefficient and mass transfer driving force [87]. There are many ways to improve the mass transfer effect, including increasing the gas–liquid mass transfer coefficient or the specific surface area [5], the intensification of agitation in the reaction vessel [79], increasing the partial pressure and flow rate of the feed gas, improving the solubility and decomposition rate of ozone (e.g., improving the reaction temperature and decreasing the size of the microbubbles), etc. [52,71,87,93,168]. It should be noted

that the volumetric mass transfer coefficient depends on the interfacial area and is directly related to the properties of activated carbon [87].

The effect of the pore structure of activated carbon catalyst on mass transfer is generally reflected in two aspects: the adsorption capacity and catalytic capacity. The former affects mass transfer by adsorbing ozone molecules from the liquid phase, while the latter influences mass transfer by increasing the concentration gradient through the rapid depletion of ozone molecules in the reaction. Akhtar et al. [161] proposed that the decomposing of dissolved ozone into free radicals would increase ozone mass transfer. The adsorption capacity was mainly affected by the pore volume and specific surface area of the adsorbent. Some studies have shown that the activated carbon with a large surface area will reduce the mass transfer resistance and encourage the adsorption of a wide range of organic matter [27,101]. Biernacki et al. [79] verified that the ozone decomposition constant increased with the increase in the specific surface area of activated carbon, and then the restricted mass transfer process can only occur between the aqueous phase and the external specific surface of activated carbon.

Generally, the more developed the pore structure of the catalyst, the larger the specific surface area, and the faster the absorption rate from the liquid phase interface to the solution body; then, the greater the concentration difference between the ozone molecules at the gas phase interface and liquid phase interface, the greater the mass transfer force, which is conducive to mass transfer. Various studies have proved this point [79,93]. In a slurry reactor, the mass-transfer-limited reaction can be significantly improved by increasing the specific surface area [5]. In the gas–liquid–solid circulating fluidized bed catalytic ozonation coal chemical industry wastewater treatment process, Li et al. [30] found that compared with other two catalysts, Mn/AC could increase the liquid–solid contact efficiency and promote mass transfer. Increasing the mass transfer rate of ozone from gas phase to liquid phase can keep the ozone concentration in the liquid phase at a low level and improve the utilization rate of ozone. However, the researcher did not explain the phenomenon that the catalyst supported by activated carbon has better ozone mass transfer efficiency and ozone utilization ratio than a catalyst supported by other carriers. Mesopores are an important factor affecting the adsorption of large size molecules, therefore activated carbon with a certain proportion of mesopores is conducive to the mass transfer and degradation of large molecules [161]. Messele et al. [53] confirmed that mesopores activated carbon has high pore volume, and wide and available pores, which is conducive to a better diffusion of ozone and organic matter, thus improving the reaction rate.

The proportion of macropores, mesopores, and micropores in the activated carbon catalysts will also affect the mass transfer process, especially since the existence of a certain proportion of mesopores is very necessary. Because the internal surface area accounts for most of the activated carbon, limited mesopores will limit the mass transfer of ozone molecules from the bulk solution to the activated carbon internal surface [64]. Chen et al. [39] obtained super fine granular activated carbon by ball milling and used it as a catalyst carrier for catalytic ozonation of oxalic acid solution. They found that an increase in the proportion of macropores and mesopores in activated carbon can greatly reduce the mass transfer resistance of oxalate to activated carbon carriers and increase the adsorption rate and reaction rate. However, the author did not explain the specific proportion of macropores and mesopores in the total pore volume and whether the conclusion is universal.

## 5. Conclusions

The pore structure characteristic of the activated carbon catalyst is one of the key factors in catalytic ozonation, which has a great influence on adsorption, catalytic efficiency, and mass transfer. These affect the catalytic efficiency by adsorption ability, catalytic active sites, and the generation of free radicals, while these in turn affect the mass transfer of ozone molecules from the gas phase interface to the liquid phase interface, and from the bulk solution to the catalyst surface. In this paper, the pore structure regulation methods of

activated carbon catalysts and the effect of pore structure on catalytic ozonation and mass transfer were reviewed. The main conclusions are as follows:

1.  The regulation method of the pore structure of activated carbon is relatively comprehensive and mature, and the adjustment mechanism of the pore structure of activated carbon is relatively clear; the main adjustment steps include the raw material, carbonization, activation, additive, and loading. Therefore, to produce activated carbon with a developed pore structure, suitable raw materials should be selected (for example biomass material), suitable carbonization and activation conditions (such as carbonization temperature, activation method, activation agent, activation temperature, activation times, etc.), and suitable modification and loading.

2.  The external specific surface of activated carbon catalyst is the main reaction site, which affects the efficiency of adsorption and free radical production. Therefore, the external specific surface area, not the internal specific surface area or total specific surface area, is the valid parameter.

3.  Reasonable pore size distribution has a great influence on catalytic ozonation and mass transfer. Micropores are the main suppliers of specific surface area, while mesopores are molecular transport channels. Without sufficient mesopores, the role of micropores cannot be fully reflected. The specific ratio of mesopores to micropores depends on different application requirements, so catalysts with appropriate pore structure characteristics should be developed according to wastewater characteristics and removal criteria.

4.  The external surface area and pore size distribution are the core of the pore structure of the activated carbon catalyst. The larger the external surface area and the more reasonable the pore size distribution, the more conducive is the activated carbon to catalytic ozonation and mass transfer. The main effect of pore structure on catalytic ozonation is to provide a catalytic active site and promote free radical generation, but the specific reaction mechanism is still unclear. The effect of pore structure on mass transfer is less studied, and it mainly reduces the mass transfer resistance.

**Author Contributions:** J.Y. conducted the research and investigation process and wrote the original draft. L.F. have made substantial contributions to the conception of the work, revising it critically for the important intellectual content. F.W. acquisition, analysis, and interpretation of data for the work. X.C. acquisition and analysis data for the work. C.W. have made substantial contributions to the conception of the design of the work; reviewed and edited the manuscript, and gave final approval for the version to be published. Q.W. reviewed and edited the manuscript. All authors have read and agreed to the published version of the manuscript.

**Funding:** This work was funded by National key R&D projects [funder: Changyong Wu] [grant number: 2020YFC1806302-03].

**Conflicts of Interest:** The authors declare no conflict of interest.

## Nomenclature

| | |
|---|---|
| AOP | Advanced oxidation processes |
| •OH | Hydroxyl radicals |
| AC | Activated carbon |
| ACF | Activated carbon fiber |
| GAC | Granular activated carbon |
| PAC | Powder activated carbon |
| SBAC | Sludge-based activated carbon |
| ROS | Reactive oxygen species |
| $O_2^{\bullet-}$ | Superoxide radical |
| $^1O_2$ | Singlet oxygen |
| $H_2O_2$ | Hydrogen peroxide |
| PAH | Polycyclic aromatic hydrocarbon |
| ADA | 1,3-Adamantanedicarboxylic acid |

| | |
|---|---|
| TOC | Total organic carbon |
| HTC | Hydrothermal carbonization |
| $S_{BET}$ | Specific surface area (BET method) |
| BET | Brunauer-Emmett-Teller |
| $V_{micro}$ | Micropore volume |
| $V_{total}$ | Total pore volume |
| $D_p$ | Average pore diameter |
| M | Moisture content |
| V | Volatile matter |
| FC | Fixed carbon |
| A | Ash |
| DBP | Dibutyl phthalate |

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
