# Peer review of "Recent Developments in Activated Carbon Catalysts Based on Pore Size Regulation in the Application of Catalytic Ozonation"

_catalysts, doi:10.3390/catal12101085_

Round 1

Reviewer 1 Report

This review paper entitled “The recent development of activated carbon catalyst based on pore size regulation in the application of catalytic ozonation: A review” summarized recent development of activated carbon catalyst in ozonation reaction. The authors also studied the pore size and structure of carbon in the reaction. The paper appropriately cited many papers in this field and is scientifically sound. I have a few concerns for the authors to address.

1. Some grammatical error has to be corrected (e.g. Line 102. which can be provide reference)

2. This paper focused on the use of activated carbon in ozonation. I would suggest the authors to include some other commonly used catalyst (e.g. zeolite) for comparison when mechanisms were illustrated or efficiencies were demonstrated. I also suggest the comparison between H2O2 and ozone in the mechanism part since they have some similarities.

3. When the pore size was discussed, the porosity and surface area are also important to study. It would be better if the authors can summarize some data in a table.

Reviewer 2 Report

The review by Jun yang et al. Entitled "The recent development of activated carbon catalyst based on pore size regulation in the application of catalytic ozonation: A review" is very interesting, novel and well-organised. However, following comments can contribute to the manuscript, positively:

1. Please check and correct the mistakes in author list, affiliations and the names of corresponding authors. 

2. The title is too long. Please make it concise.

3. Please get the manuscript checked for grammatical errors.

4. Reference style must be aligned as per the desired journal.

5. Please improve the last paragraph of Introduction.

6. Some content of the figure 4 is not visible (right side), please correct.

7. In Table 1, please add "-" in the cells, whose values are not available.

8. Please add a list of abbreviations.

9. Please cite some latest literature on activated carbon and wastewater treatment: Journal of Environmental Chemical Engineering 8 (5), 104220, 2020; Chemosphere, 135566, 2022; Journal of Environmental Chemical Engineering 7 (4), 103265, 2019; Environmental Science and Pollution Research 28, 9050–9066, 2021. 
